# A Theoretical Framework for Implementable Nucleic Acids Feedback Systems

**DOI:** 10.3390/bioengineering10040466

**Published:** 2023-04-12

**Authors:** Nuno M. G. Paulino, Mathias Foo, Tom F. A. de Greef, Jongmin Kim, Declan G. Bates

**Affiliations:** 1School of Engineering, University of Warwick, Coventry CV4 7AL, UK; 2Department of Biomedical Engineering, Eindhoven University of Technology, 5600 MB Eindhoven, The Netherlands; 3Department of Life Sciences, Pohang University of Science and Technology (POSTECH), Pohang 37673, Gyeongbuk, Republic of Korea

**Keywords:** feedback control, chemical reaction networks, nucleic acids, strand displacement circuits, synthetic biology

## Abstract

Chemical reaction networks can be utilised as basic components for nucleic acid feedback control systems’ design for Synthetic Biology application. DNA hybridisation and programmed strand-displacement reactions are effective primitives for implementation. However, the experimental validation and scale-up of nucleic acid control systems are still considerably falling behind their theoretical designs. To aid with the progress heading into experimental implementations, we provide here chemical reaction networks that represent two fundamental classes of linear controllers: integral and static negative state feedback. We reduced the complexity of the networks by finding designs with fewer reactions and chemical species, to take account of the limits of current experimental capabilities and mitigate issues pertaining to crosstalk and leakage, along with toehold sequence design. The supplied control circuits are quintessential candidates for the first experimental validations of nucleic acid controllers, since they have a number of parameters, species, and reactions small enough for viable experimentation with current technical capabilities, but still represent challenging feedback control systems. They are also well suited to further theoretical analysis to verify results on the stability, performance, and robustness of this important new class of control systems.

## 1. Introduction

With the recent advancement of cost-effective approaches for assembling oligonucleotides, synthetic DNA has become an appealing preference for constructing circuits using hybridisation and DNA strand displacement (DSD) reactions. Programmable and biologically compatible synthetic circuits that utilise DSD reactions have been shown to function effectively in living organisms (in vivo), as reported in a study by [1]. Therefore, these DSD networks are a promising candidate for building synthetic feedback controllers in biomolecular settings [2], to drive DNA-based molecular devices, or for the realisation of biochemical computing circuits with emerging opportunities and challenges also for applications in vivo [3].

Systematic methods to compose and program DNA circuits have been proposed [4] that enable the translation of chemical programs to reactions through the use of nucleic acids, where the binding affinities and the reaction rates can, to a certain degree, be controlled [5,6]. Feedback control systems can be assembled with elementary chemical reaction networks (CRNs) that provide an abstract layer for designing mathematical operators [7], prior to being translated to biomolecular applications. As a result, the networks of fundamental reactions such as catalysis, degradation, and annihilation can then be mapped to their equivalent DSD reactions [8], where the DNA sequences effectively program the biochemical interactions and each of the strand displacement reaction rates [5,9].

Nonetheless, at first glance, the concentrations of biomolecular species are seemingly ill-suited to represent signals in the theory of feedback control, given that they are limited to positive quantities. For instance, in a typical reference tracking control problem, the error between the reference and the plant output can take either a negative or a positive value, but then, this cannot be represented by a positive concentration. An alternative technique that works around this limitation is to employ the so-called dual-rail representation, in which pairs of concentrations are used to represent system state variables [10]. Specifically, a signal x(t) is described by the difference between two concentrations $x\left(t\right)=x^{+}\left(t\right)-x^{-}\left(t\right)$, which corresponds to two chemical species $X^{+}$ and $X^{-}$. By making such an abstraction, this approach enables the representation of gains and negative signals using positive quantities that are essential for error computation in a negative feedback control system. The dual-rail representation and implementations for elementary catalysis, degradation and annihilation reactions provide a systematic framework to convert control theory into biochemistry, which can be implemented with synthetic DNA oligonucleotides, where the representation of transfer functions [11], linear feedback systems [12,13,14,15,16,17], and nonlinear controllers [18,19,20] can be realised through the assembly of DSD networks.

While open-loop cascades for analogue or logical computations implemented with DSD networks have been constructed [21], there remains a considerable gap between theory and experiment when it comes to dynamical circuits that implement negative feedback. Moreover, even the most basic types of feedback controllers have yet to be implemented and reported. This is an important limitation to further progress, because the proposed dual-rail representation requires annihilation reactions, which are essential to ensure that species concentrations remain within the bounds of experimental feasibility, but lead to nonlinear dynamics due to the bimolecular nature of annihilation reactions.

Ideally, these additional nonlinear dynamics are not observable in the represented input to output linear dynamics, but still determine some important properties of the CRN, which need careful consideration for experimental implementation, such as the level of concentrations at which the circuit operates and unforced positive equilibria with persistent consumption of chemical species [22,23]. Additionally, the implementation of the DSD reactions introduce leakage and spurious effects [24], which prompted study on leakage analysis and ways to mitigate it [25]. Furthermore, the variability and granular design of hybridisation rates [5] introduce error and mismatches in the reaction rates. All these aforementioned issues show that we are lacking an experimental validation of control systems based on DNA hybridisation, to help us understand how nonlinearities and uncertainty in the implementation impact the stability and performance of these control circuits.

Due to several technical challenges, which include cross-talk, designing toehold sequences, leakage, etc., there is a substantial lag in the construction, validation, and scale-up of nucleic acid control systems compared to the theoretical advancements’ counterpart [5,24]. While the nucleotide sequence can be predicted from the toehold affinities [5], the kinetics can be affected by undesirable binding scenarios, resulting in the modification of the reaction rate constants and the dynamics (e.g., in oscillating circuits [24] or the seesaw gate circuit [26]). The activation of these undesirable reactions that yields the leakage of the outputs in the absence of the input also poses another major experimental hurdles. A potential approach in addressing this issue is the use of clamps, which could impede the spurious hybridisations [24,27], or compartmentalisation and localisation strategies [28,29,30], which isolate strand complexes, which could cause leak reactions. Additionally, the severity of leakage increases when concentrations are high; thus, it is a common practice to ensure low concentrations of the reacting species [31]. This leakage at high concentrations along with the restrictions on hybridisation rates [5] impose upper bounds on the speed at which these circuits can operate. A technique that can be used to mitigate this is localisation, whereby the adjacent gates are placed for interaction without diffusion at faster rates [26]. Lastly, despite measures taken to prevent these spurious reactions, it is still essential that the sequence of reactions, which are competing for common reactants, are well-managed through the means of timescale separation or compartmentalisation [32]. With larger and more complex circuits, sequence similarity aggravates all these undesired effects, with a stronger need for toehold-mediated strand displacement circuits, which are robust to molecular noise [33].

All the experimental challenges described above can be addressed through the designs of circuits that have a lesser amount of reactions. With a lesser amount of reactions and species, this reduces the number of designs required for the template strands. This concurrently reduces the demand for the characterisation and tuning of the kinetic rates, lessens the sources of error, as well as curtails the possibility of leakage and undesired interactions. Ensuring fewer reactions and species are of pertinence, especially in using the dual-rail representation, which necessitates a duplication of nearly all of the reactions [19].

A key challenge from a theoretical perspective is, therefore, to design circuits of reduced complexity, using the least amount of chemical reactions, at the same time still able to represent a negative feedback control problem of interest in an accurate manner. All these are done to ensure that, with the presently available technical capacity, the likelihood of implementing a successful experiment can be maximised. Here, we suggest chemical representations for two fundamental types of feedback control systems in order to achieve the above objective. Our first type of circuit is a reference-tracking problem employing integral control for a stable first-order system, with a single tuning parameter. The second type of circuit is a combined reference tracking and stabilisation problem of a marginally stable plant, i.e., the classic double-integrator system employing the static state feedback control of the two integrated states with two tunable gains. The dual-rail framework is used in both of these circuits, and their parametrisations can be tuned to satisfy control requirements such as steady-state tracking and transient dynamics. The supplied circuits of reduced complexity are ideal candidates for the first experimental validation and robustness assessment of nucleic acid controllers implemented via the dual-rail representation.

An early version of this study can be found in [34], where we established the fundamental theories of chemical reaction network that are required to realise the reference tracking control problem. Here, we extended the work in [34] by applying the theories to design two types of controllers and investigated the impact of the nonlinearities and uncertainties on the experimental implementation, in particular the effect of leakage, which was not considered previously.

The remainder of this article is organised as follows. Section 2 introduces the definitions and methods used in this work. The formulations and constructions with DNA hybridisation reactions are detailed for control systems with integral action in Section 3 and with state feedback in Section 4. Section 5 discusses the impact of nonlinearities and uncertainty in performance and robustness, highlighting the properties of interest for the experimental investigation. Section 6 summarises the main conclusions.

## 2. Definitions and Preliminaries

### 2.1. Chemical Reaction Networks

The theory of CRNs has its roots in chemistry [35], whereby the same principle is applied in a similar manner to biomolecules to model and analyse biochemical reaction networks. With their extensive computational capabilities [36,37], CRNs provide a convenient representation for implementing elementary arithmetic operations [7] or the computation of polynomials [38], using any chemical system with mass action kinetics. They also provide an appropriate level of abstraction for designing complex circuits [39] and integrate the different elements necessary to build linear feedback control systems [12,13].

**Definition** **1.**
*Given a set of M reactions between N chemical species $X_{j}$, a chemical reaction network is represented by*

(1)
$$\begin{array}{c} \sum_{j=1}^{N}a_{jm}X_{j}\overset{\gamma_{m}}{\to }\sum_{j=1}^{N}b_{jm}X_{j},\;\left(m=1,\ldots ,M\right) \end{array}$$

*where the stoichiometric coefficients $a_{jm}$ and $b_{jm}$ determine, for each reaction m, the quantities of the reactants consumed on the left and the products output on the right, at a rate $\gamma_{m}$. When, due to degradation or sequestration, a product is removed or becomes nonreactive, it is omitted or replaced by the symbol *Ø*.*


When the same species is utilised in multiple reactions, it creates links between different species that can be expressed as a *graph* to describe the network. There are various sources that provide the definitions of the models and properties pertaining to CRNs. These include Petri nets and monotone systems [40,41]; deficiency theorems and structural analysis work [42,43,44,45]; and biochemical networks’ application [46].

Different principles have been used to model the time evolution of the concentration of species [35]. In this study, the main principle that we used is the Mass-Action-Kinetics (MAK)-based deterministic models, which have been traditionally used to model biochemical processes [47,48,49], as well as used in the implementation of polynomial Ordinary Differential Equations (ODEs) and Turing universal CRNs [36,50]. Assuming the MAK, a set of ODE can be used to express a deterministic model [51,52,53] that is appropriate for the analysis of control and systems theory, despite being nonlinear, with the reaction fluxes depending on the product of the reatanct’s concentrations.

**Definition** **2.**
*For a deterministic model based on the MAK [35], we have that, for each species $X_{j}$ in (Equation 1), the dynamics of its concentration $x_{j}$ is given by*

(2)
$$\begin{array}{c} {\dot{x}}_{j}=\sum_{m=1}^{M}\left(\gamma_{m}\left(b_{jm}-a_{jm}\right)\prod_{j=1}^{N}x_{j}^{a_{jm}}\right) \end{array}$$



**Example** **1.**
*Take the reaction between the reactant species $X_{1}$ and $X_{2}$, which produces species $X_{3}$, represented by*

(3)
$$\begin{array}{c} a_{1}X_{1}+a_{2}X_{2}\overset{\gamma }{\to }bX_{3} \end{array}$$

*where the reactants on the left are converted into the product on the right at a rate γ, according to the stoichiometric coefficients $a_{1}$, $a_{2}$, and b. The MAK model results in the ODEs:*

(4)
$${\dot{x}}_{1}=-a_{1}\gamma x_{1}^{a_{1}}x_{2}^{a_{2}}$$


(5)
$${\dot{x}}_{2}=-a_{2}\gamma x_{1}^{a_{1}}x_{2}^{a_{2}}$$


(6)
$${\dot{x}}_{3}=+b\gamma x_{1}^{a_{1}}x_{2}^{a_{2}}$$

*where the stoichiometric coefficients $a_{1}$, $a_{2}$, and b indicate, respectively, the relative number of molecules consumed and produced during the reaction.*


Let $R_{0}^{+}$ denote the positive orthant, where $v_{j}\in R_{0}^{+}\Rightarrow v_{j}\geq 0$; the dynamics of (Equation 4) to (Equation 6) are expressed in their natural coordinates, and the concentrations $x_{j}\in R_{0}^{+}$ are always non-negative. The system is non-negative ($\forall_{j},x_{j}\left(0\right)\geq 0\Rightarrow x_{j}\left(t\right)\geq 0$) if the sets of reactions are modelled properly using realistic non-negative initial conditions.

### 2.2. Dual-Rail Representation

In the context of feedback control, a key challenge in employing CRNs is their inability to directly represent negative signals, since the concentrations of chemical species are always positive. In a linear negative feedback system, we cannot restrict ourselves to considering only a positive difference between two positive inputs, i.e., “one-sided” subtraction [54]. The limitation of the positivity of CRNs can be circumvented using the dual-rail representation [10,12,55], where this enables the positive and, in particular, the negative signals that are vital for error signals’ computation in a feedback control to be represented using molecular concentrations.

**Definition** **3.**
*Consider two chemical species $X_{j}^{+}$ and $X_{j}^{-}$ and respective concentrations $x_{j}^{+}\geq 0$ and $x_{j}^{-}\geq 0$. A dual-rail signal $p_{j}\in R$ is represented by $p_{j}=x_{j}^{+}-x_{j}^{-}$, with dynamics given by ${\dot{p}}_{j}={\dot{x}}_{j}^{+}-{\dot{x}}_{j}^{-}$.*


Here, we have signals that are generated by subtracting two positive quantities, which enables us to obtain positive or negative results when subtracting signals instead of concentrations.

**Definition** **4.**
*The Input–Output (I/O) system is the response Y(s)=G(s)U(s), from an input $u=\left(u^{+}-u^{-}\right)$ to an output $y=\left(y^{+}-y^{-}\right)$. The states are also dual-rail $p_{j}=x_{j}^{+}-x_{j}^{-}$, where $u,y,p_{j}\in R$ and $u^{\pm },y^{\pm },x_{j}^{\pm }\in R_{0}^{+}$.*


Combining the MAK and the dual-rail representation, we can model linear systems with both positive and negative signals, using CRNs composed of only the three types of reactions:(7)$$X_{i}\overset{\gamma }{\to }X_{i}+X_{j}$$
(8)$$X_{i}\overset{\gamma }{\to }Ø$$
(9)$$X_{i}+X_{j}\overset{\eta }{\to }Ø$$
where we have unimolecular reactions for catalysis (Equation 7) and degradation (Equation 8) and a bimolecular reaction in annihilation (Equation 9).

**Example** **2**(First-order system)**.** *To chemically represent the first-order system with a negative gain given by the transfer function:*
(10)$$\begin{array}{c} Y\left(s\right)=-\frac{k_{1}}{s+k_{2}}U\left(s\right) \end{array}$$
*with u,y∈R, we take the pairs of chemical species $\left\{U^{\pm },Y^{\pm }\right\}$ and the CRN from Figure 1 with*
(11)$$U^{\pm }\overset{k_{1}}{\to }U^{\pm }+Y^{\mp }$$
(12)$$Y^{\pm }\overset{k_{2}}{\to }Ø$$
(13)$$Y^{+}+Y^{-}\overset{\eta }{\to }Ø$$
*From the MAK, we derive the ODEs for the concentrations:*
(14)$${\dot{y}}^{+}=-k_{2}y^{+}+k_{1}u^{-}-\eta y^{+}y^{-}$$
(15)$${\dot{y}}^{-}=-k_{2}y^{-}+k_{1}u^{+}-\eta y^{+}y^{-}$$
*From Definition 4, the linear I/O system results:*
(16)$$\begin{array}{ccc} {\dot{y}}^{+}-{\dot{y}}^{-} & = & -k_{2}\left(y^{+}-y^{-}\right)-k_{1}\left(u^{+}-u^{-}\right)\Leftrightarrow \dot{y}=-k_{2}y-k_{1}u \end{array}$$

The computation of a two-sided subtraction with the steady-state of a CRN or the representation of negative gains is enabled by the dual-rail representation [55]. Although this increases the number of required reactions (by duplicating the unimolecular reactions), it does provide a systematic framework with which to represent linear and negative feedback systems [11,12,14].

### 2.3. DNA Strand Displacement Reactions

Sequences of four types of nucleotides, i.e., Adenine (A), Guanine (G), Cytosine (C), and Thymine (T), are formed by chains of nucleotides in nucleic acids. The formation of hydrogen bonds between nucleotide pairs A-T and C-G results in a double-stranded DNA with a helical shape. As depicted in Figure 2, this DNA structure is composed of two strands of DNA that run antiparallel to each other. The bonding between the complementary pairs of A-T and C-G is stabilised by the hydrophilic backbone and the hydrophobic nature of the bases. As illustrated in Figure 2b, this results in the formation of double-stranded DNA through an enzyme-free hybridisation reaction between two antiparallel complementary strands of DNA [56].

The dynamics of simple chemical reactions such as catalysis, degradation, and annihilation can be readily translated into nucleic-acid-based reactions with comparable behaviours [8,9]. This approach enables technology for implementing circuits designed using CRNs, where the DNA sequence of the nucleotides within the strands can efficaciously be used to program the biochemical circuitry to realise both analogue and digital computations [4,5,26,54]. Each of these reactions has equivalent representations with DSD reactions, where formal species and reactions in the CRN are mapped to sets of DNA species. Various suggested architectures [8,9,24,32] permit the creation of circuits that are aided by a high-level of automation through the utilisation of existing syntax and software tools [4,13,57].

The DSD reaction uses DNA molecules as signal species, and they consist of binding domains and single-stranded overhanging and exposed toeholds. The function of these toeholds is to initiate and adjust the reaction rates of strand hybridisation [4,5], as depicted in Figure 2c. The hybridisation process leads to bimolecular reactions with unitary stoichiometric coefficients. Using the following DSD reaction of A+B→C+D as an illustration (Figure 3), the toehold 1 of the approaching strand *A* is hybridised to its complementary toehold $1^{*}$ in *B*, which leads to the commencement of branch migration. This then displaces Domain 2 and releases the two output species of *C* and *D*. From the thermodynamic point of view, the unbinding of toehold 1 is less favoured compared to the displacement of the incumbent domain 2, resulting in *C* being completely displaced. With the two output species *C* and *D* having no overhanging complementary toeholds for further hybridisation, the displacement mechanism is deemed not reversible, assuming leakage reaction rates are much slower than the displacement mediated by the toehold reactions. The reverse reaction, where *C* displaces the strand *A* from *D*, can be up to orders of magnitude slower [58] and be considered effectively unreactive.

With the assumption of an irreversible reaction coupled with the output strand *C* being able to take part in other reactions, the cascade of several reactions ensues. It is often a challenge to describe the hybridisation reaction with good accuracy. Nonetheless, there are several predictive models available that utilise the nucleotide sequences to provide a good approximation of the reaction rate [5]. Using Visual DSD [57], which is a specifically designed computer-aided design software for strand displacement reactions’ analysis, we conducted a simulation to analyse and verify the DSD circuity by programming the species and their affinities in the software. We ran the simulation in “default” mode, which is the mode that considers finite binding/unbinding rates. The affinities between the toeholds, which facilitate the initial hybridisation between the strands, are determined based on the reaction rates in the CRNs.

## 3. Integral Feedback Control System

We now propose a CRN with reduced complexity to represent the integral feedback control system in Figure 4. This is then followed by implementing it in Visual DSD using strand displacement reactions.

**Example** **3**(Reference tracking problem using integral control)**.** *Define the first-order plant and the control law:*
(17)$$\begin{array}{ccc} Y(s) & = & \frac{b}{s+a}V\left(s\right)\Rightarrow sY\left(s\right)=bV\left(s\right)-aY\left(s\right) \end{array}$$
(18)$$\begin{array}{ccc} V(s) & = & \frac{k_{i}}{s}\left(R(s)-Y(s)\right)\; \end{array}$$
*where V(s) is the integral control action to have the output Y(s) track a reference R(s), according to Figure 4.*

The transfer function for the closed-loop dynamics results:(19)$$\begin{array}{c} Y\left(s\right)=\frac{bk_{i}}{s^{2}+as+bk_{i}}R\left(s\right) \end{array}$$
where the integral control ensures steady-state tracking with Y(0)=R(0). The transient dynamics are defined by the roots of the characteristic polynomial:(20)$$\begin{array}{c} \lambda =\frac{1}{2}\left(-a\pm \sqrt{a^{2}-4bk_{i}}\right) \end{array}$$
with a natural frequency $\omega_{n}=\sqrt{bk_{i}}$ and damping coefficient $\xi =\frac{a}{2\sqrt{bk_{i}}}$.

### 3.1. Representation with Chemical Reactions

The closed-loop response represented by a network of catalysis, degradation, and annihilation reactions is shown in Figure 5, where
(21)$$V^{+}\overset{b}{\to }V^{+}+Y^{+},\;\;V^{-}\overset{b}{\to }V^{-}+Y^{-}$$
(22)$$Y^{+}\overset{a}{\to }Ø,\;\;Y^{-}\overset{a}{\to }Ø$$
(23)$$R^{+}\overset{k_{i}}{\to }R^{+}+V^{+},\;\;R^{-}\overset{k_{i}}{\to }R^{-}+V^{-}$$
(24)$$Y^{+}\overset{k_{i}}{\to }Y^{+}+V^{-},\;\;Y^{-}\overset{k_{i}}{\to }Y^{-}+V^{+}$$
(25)$$Y^{+}+Y^{-}\overset{\eta }{\to }Ø,\;\;V^{+}+V^{-}\overset{\eta }{\to }Ø$$

The reactions (Equation 21) and (Equation 22) describe the plant, where the reaction rates *a* and *b*, respectively, set the gain and the degradation reactions, with a stable pole at s=−a.

The additional dynamics of the chemical reactions used to solve the subtraction and compute the control error as found in [12] or [14] are removed. Instead of using separate CRNs for subtraction and integration, both operations are combined to reduce the number of reactions and simplify the circuit. Thus, Equations (Equation 23) and (Equation 24) describe the subtraction of the contributing reference and the output to the integral action through the crossing of the contributions from $Y^{\pm }$ to $V^{\mp }$. The control gain is applied by the reaction rate $k_{i}$ in the corresponding catalysis reactions. The annihilation reactions in (Equation 25) ensure low (i.e., experimentally feasible) concentrations of the chemical species [12,23].

The respective MAK governing the species concentrations can be written as
(26)$${\dot{v}}^{+}=k_{i}r^{+}+k_{i}y^{-}-\eta v^{+}v^{-}$$
(27)$${\dot{v}}^{-}=k_{i}r^{-}+k_{i}y^{+}-\eta v^{+}v^{-}$$
(28)$${\dot{y}}^{+}=bv^{+}-ay^{+}-\eta y^{+}y^{-}$$
(29)$${\dot{y}}^{-}=bv^{-}-ay^{-}-\eta y^{+}y^{-}$$

This gives us then that, for the dual-rail variables $y=y^{+}-y^{-}$, $v=v^{+}-v^{-}$, and $r=r^{+}-r^{-}$, the dynamics of the I/O are the outcome of the expressions $\dot{v}={\dot{v}}^{+}-{\dot{v}}^{-}$ and $\dot{y}={\dot{y}}^{+}-{\dot{y}}^{-}$, and this yields
(30)$$\dot{v}=k_{i}\left(r^{+}-r^{-}\right)+k_{i}\left(-y^{+}+y^{-}\right)=k_{i}\left(r-y\right)$$
(31)$$\dot{y}=bv-ay$$

The swapped contributions from $Y^{\pm }$ to $V^{\mp }$ still result in positive gains in the dynamics of the concentrations in (Equation 26) to (Equation 29), but result in a negative gain on the output *y* when computing the dynamics in (Equation 30) to (Equation 31).

The dynamics of the input–output given in (Equation 30) to (Equation 31) are linear, corresponding to the transfer function of the closed-loop system in (Equation 19). To further simplify the construction, we disregarded any annihilation reaction among the reference species $R^{+}+R^{-}\overset{\eta }{\to }Ø$ with the assumption that we have low and constant input concentrations during the whole circuit operation, and the dynamics in (Equation 30) to (Equation 31) rely solely on the difference of *r* instead of the $R^{+}$ and $R^{-}$ concentration levels.

#### 3.1.1. Dual-Rail CRN as a Nonlinear Internal Positive Representation

The dual-rail representation uses a positive system (the MAK of the CRN) to represent a non-positive system. This approach of having an Internally Positive Representation (IPR) [59] lifts the constraint of positivity imposed by the implementation of the physical system using chemical species. The representation uses input and output transformations that translate the input and output of the system into and from positive states of the dynamics, respectively, to realise arbitrary I/O dynamics. In our case, the internal positive dynamics are represented by CRNs (Figure 6) and the positive variables are the natural coordinates of the MAK given by positive concentrations.

When representing linear I/O systems with a linear IPR, we need to ensure that the system is stable from input to output, but also internally stable. This results in limitations when building the internal linear process [59]. However, the dual-rail representation constructed with the elementary reactions of catalysis, degradation, and annihilation reactions results in a nonlinear MAK. The I/O system in (Equation 30) to (Equation 31) is linear because the dynamics of the differences of concentrations have the nonlinear terms of the MAK in (Equation 26) to (Equation 29) cancelling out. However, internally, we have a process that is not only positive, but also nonlinear and not observable in the I/O system.

**Definition** **5**(Rotated coordinates [23])**.** *Define the transformation of coordinates $\bar{v}=v^{+}-v^{-}$ and $\hat{v}=v^{+}+v^{-}$, where we can recover the concentrations with $v^{\pm }=\frac{1}{2}\left(\hat{v}\pm \bar{v}\right)$.*

In the new coordinates, we can clearly express the relationship between the I/O system and the internal positive nonlinear dynamics [23]. Since $\eta y^{+}y^{-}=\frac{\eta }{4}\left({\hat{y}}^{2}-{\bar{y}}^{2}\right)$, we can express the MAK of Example 3 in the new coordinates, where we obtain
(32)$$\dot{\bar{v}}=k_{i}\left(\bar{r}-\bar{y}\right)$$
(33)$$\dot{\bar{y}}=b\bar{v}-a\bar{y}$$
(34)$$\dot{\hat{v}}=k_{i}\left(\hat{r}+\hat{y}\right)-\frac{\eta }{2}{\hat{v}}^{2}+\frac{\eta }{2}{\bar{v}}^{2}$$
(35)$$\dot{\hat{y}}=b\hat{v}+a\hat{y}-\frac{\eta }{2}{\hat{y}}^{2}+\frac{\eta }{2}{\bar{y}}^{2}$$

The dynamics of $\dot{\bar{v}}$ and $\dot{\bar{y}}$ correspond to the I/O system and $\dot{\hat{v}}$ and $\dot{\hat{y}}$ are the remaining dynamics in the dual-rail representation, which are nonlinear and depend on the trajectories of the I/O system. It is noteworthy that the I/O system does not rely on $\dot{\hat{v}}$ or $\dot{\hat{y}}$, and these underlying nonlinear dynamics are, by design, not observable in the represented linear system.

**Assumption** **1.**
*It was assumed that the designed I/O system results in a stable closed feedback system.*


Assuming that the controller is designed to ensure stability for the feedback system, we have that the I/O system is a bounded and stable input into the internal nonlinear dynamics. Assumption 1 together with previous theoretical results from [23] ensure that, although nonlinear and partially unobservable, the dynamics in (Equation 32) to (Equation 35) remain stable.

### 3.2. Representation with Strand Displacement Reactions

In order for CRNs to be translated into nucleic acid chemistry, the DSD reactions are set with comparable dynamics to each of the three elementary reactions, coupled with toehold mediation to tune their reaction rates. Following [9], both catalysis and degradation reactions can be implemented using the Join–Fork templates. A signal strand can be released through the interaction of a Join double-stranded complex with either one or a few single-stranded inputs. This then leads to a sequence of DSD reactions in the Fork template, where one or more output strands are released. The auxiliary templates, namely the Join and Fork templates, as well as the auxiliary two-domain strands, need to be present in high concentrations and supplied at the start of the reaction.

Figure 7 shows an example of sets of DSD reactions equivalent to the unimolecular reaction of catalysis $R\overset{}{\to }R+V$, where the interactions with the intermediary strands and auxiliary templates are mediated utilising toeholds. The sets of reactions are
(36)$$R+\mathrm{JoinVR}\overset{c_{i}\times k_{t}}{\underset{k_{bnd}}{⇌}}JoinVR\_1+\mathrm{aux}\_\mathrm{hrtp}$$
(37)$$JoinVR\_1+\mathrm{aux}\_\mathrm{tphpr}\overset{k_{bnd}}{\underset{k_{bnd}}{⇌}}JoinVR\_2+sig\_hprtq$$
(38)$$sig\_hprtq+\mathrm{ForkVR}\overset{k_{bnd}}{\underset{k_{bnd}}{⇌}}ForkVR\_1+\mathrm{aux}\_\mathrm{tphpr}$$
(39)$$ForkVR\_1+\mathrm{aux}\_\mathrm{hrtp}\overset{k_{bnd}}{\underset{k_{t}}{⇌}}ForkVR\_2+R$$
(40)$$ForkVR\_2+\mathrm{aux}\_\mathrm{hvtr}\overset{k_{t}}{\underset{k_{t}}{⇌}}ForkVR\_3+V$$
(41)$$ForkVR\_3+\mathrm{aux}\_\mathrm{hitv}\overset{k_{t}}{\to }Ø$$

The species highlighted in bold in (Equation 36) to (Equation 41) are the template complexes and auxiliary single-stranded species, which are made available at a high concentration $C_{max}$, to avoid their irreversible consumption from significantly impacting the dynamics.

As shown in Figure 7, every signal species is a single-stranded DNA that consists of toehold (<tr>) and binding (<hr>) domains. The hybridisation to the multi-stranded complex JoinVR is initiated by the toehold domain, which activates a series of processes, leading to an intermediary strand sig_hprtq being released. The presence of this new intermediary strand activates another series of strand displacements, but this time involving the ForkVR complex. This results in the signal species *V* being released and a strand of *R* being returned (following the stoichiometry of $R\overset{}{\to }R+V$).

**Remark** **1.**
*The reversible reactions in *(Equation 37)* and *(Equation 38)* assume the same forward and backward binding rate $k_{bnd}$. Implicit in this assumption is that the sequences of the toeholds <tp> and <tp*> can be designed to have the same maximum binding rate as the toeholds <tq> and <tq*>. The same assumption is made in *(Equation 40)* for <tr>, <tr*>, <tv>, and <tv*>.*


As depicted in Figure 8, the DSD reactions for the degradation reaction are less complicated given that the JoinY complex only requires capturing the signal species *Y* in an irreversible manner, with
(42)$$\begin{array}{c} \mathrm{JoinY}+Y\overset{k_{t}\times c_{a}}{\to }JoinY\_1+dY \end{array}$$

Considering that there are no exposed toeholds in the product of the double-strand, this product becomes inactive and participates no further in the reactions.

To implement the annihilation reactions, we propose the cooperative hybridisation from [61] and applied in [32]. An auxiliary species AnnVV at high concentrations quickly hybridises with both input signals to generate waste species, and in order for the cascade to become irreversible, the two input species need to be present. In the ensuing cooperative hybridisation mechanism, a single template per signal is then used to set the annihilation reactions $V+V^{\prime }\overset{}{\to }Ø$, where
(43)$$\begin{array}{c} V+V^{\prime }+AnnVV\overset{c_{VV}\times k_{t}}{\to }Ø \end{array}$$

As shown in Figure 9, there is a set of DSD reactions that includes intermediary complexes together with reversible displacement reactions, resulting in the following CRNs:(44)$$\mathrm{AnnVV}+V\overset{c_{VV}\times k_{t}}{\underset{k_{ubnd}}{⇌}}Iv$$
(45)$$Iv+V^{\prime }\overset{c_{VV}\times k_{t}}{\to }Wv+Wv^{\prime }$$
(46)$$\mathrm{AnnVV}+V^{\prime }\overset{c_{VV}\times k_{t}}{\underset{k_{ubnd}}{⇌}}Iv^{\prime }$$
(47)$$Iv^{\prime }+V\overset{c_{VV}\times k_{t}}{\to }Wv+Wv^{\prime }$$

Both *V* and $V^{\prime }$, when present, facilitate the production of two double-stranded waste complexes Wv and $Wv^{\prime }$ from the second irreversible reaction.

The nucleotide sequences in the toeholds, which initiate strand hybridisation, allow for the programmability of the DSD reactions. It is well known that the predictable hybridisation kinetics is defined by the affinities between the base pairs [5] despite several factors that could likewise impact the effective reaction rate constants [24]. To facilitate our programming and analysis using Visual DSD, we adjusted the rates of the DSD reactions through the assignment of complementarity degrees between toeholds [13,57] based on the assumption that the binding affinities are weakened by the possibility of toehold design and nucleotide sequences mismatch. As illustrated in Figure 7, the complex JoinVR has the toehold <tr*ci> that possesses the complementarity degree to the signal toehold <tr> of $0<c_{i}\leq 1$. This slows down the reaction mediated by <tr> and <tr*ci> to $k_{t}\times c_{i}$, where $k_{t}$ is the maximum binding rate of two complementary toeholds <tr> and <tr*>.

**Assumption** **2.**
*The auxiliary strands were assumed to be present at high concentrations and to remain constant or approximately close to their initial value during the operation of the circuit (or indefinitely through replenishment).*


Assumption 2 allowed us to approximate the bimolecular hybridisation reactions with the unimolecular reactions in (Equation 7) and (Equation 8). For example, the implementation in (Equation 42) can be approximated assuming $\left[JoinY\right]\left(t\right)\approx \left[JoinY\right]\left(0\right)=C_{max}$, resulting in the approximating unimolecular reaction:(48)$$\begin{array}{c} Y\overset{C_{max}\times c_{a}\times k_{t}}{\to }JoinY\_1+dY \end{array}$$
where the constant and large presence of the auxiliary species JoinY is translated directly into the unimolecular reaction rate [8].

### 3.3. Initialisation and Modelling of the Integral Control System

The six catalysis and two degradation reactions from the CRN in (Equation 21) to (Equation 25) with the exception of one annihilation reaction were used in implementing the DSD reactions. In the analysis using Visual DSD, the reaction $Y^{+}+Y^{-}\overset{}{\to }Ø$ can be removed to further reduce the implementation given that both the $Y^{+}$ and $Y^{-}$ concentrations are sufficiently low.

The templates shown in Figure 7, Figure 8 and Figure 9 were employed to set the DSD reactions. A single Fork template was used for both the outputs of the catalysis reaction, where through cooperative hybridisation, the identical sequestering template is shared between the two annihilated species. A high concentration of 15 double-stranded and 20 single-stranded DNA templates and auxiliary species, respectively, was used for circuit initialisation. The domains for signal species V, V′, Y, Y′, and R, R′ that, respectively, represent $V^{+}$, $V^{-}$, $Y^{+}$, $Y^{-}$, and $R^{+}$, $R^{-}$ are shown in Figure 10. The same figure also shows the provided auxiliary templates with the details of their toeholds and domains that were constructed for hybridisation and strand displacement.

In the Visual DSD simulation, we set $C_{max}=10^{4}$ nM (the consumption of the auxiliary strands are not reversible and replenished). We also set the maximum binding and unbinding rate of the toehold to $10^{-3}\;{\left(\mathrm{nMs}\right)}^{-1}$ and $0.1\;s^{-1}$, respectively. Table 1 provides the details of all the parameters.

From Figure 11, we note that there was an agreement between the output $y=y^{+}-y^{-}$ reference tracking behaviour, as well as the concentration evolution in the DSD reactions and the CRN concentrations obtained by the MAK, as in (Equation 26) to (Equation 29). Despite the DSD reactions circuit exhibiting a desirable tracking behaviour, the dynamics of the transient were damped further. One probable reason for this is the additional auxiliary species and bimolecular reactions that were present.

## 4. State Feedback Control System

The closed-loop dynamics of the static state feedback can be modified by the controller utilising only the gains on the plant state and adding no dynamics to the open-loop system. The plant considered here was the classic double-integrator that represents the simplest second-order system. Thus, there was an extra state for the feedback apart from the output feedback.

Compared to the previous example, the control of this plant was more difficult as it was a marginally stable system due to the presence of two poles at the origin. Hence, the closed-loop system not only had to achieve the reference tracking capability, but to stabilise the open-loop system as well.

**Example** **4**(Double-integrator with linear state feedback)**.** *The state space description of the process to be controlled is*
(49)$$\begin{array}{cc} \left[\begin{array}{c} \dot{x}\\ \dot{y} \end{array}\right] & =\left[\begin{array}{cc} 0 & 0\\ q & 0 \end{array}\right]\left[\begin{array}{c} x\\ y \end{array}\right]+\left[\begin{array}{c} q\\ 0 \end{array}\right]r \end{array}$$
*where each integration has a gain q (Figure 12), and reference tracking was achieved with the control law:*
(50)$$\begin{array}{c} v=r-k_{1}x-k_{2}y \end{array}$$

There are two parameters, $k_{1}$ and $k_{2}$, that can be used to adjust the closed-loop state space system dynamics:(51)$$\begin{array}{cc} \left[\begin{array}{c} \dot{x}\\ \dot{y} \end{array}\right] & =\left[\begin{array}{cc} -qk_{1} & -qk_{2}\\ q & 0 \end{array}\right]\left[\begin{array}{c} x\\ y \end{array}\right]+\left[\begin{array}{c} q\\ 0 \end{array}\right]r \end{array}$$

The closed-loop frequency response results in a second-order system with the transfer function:(52)$$\begin{array}{c} Y\left(s\right)=\frac{q^{2}}{s^{2}+qk_{1}s+q^{2}k_{2}}R\left(s\right) \end{array}$$
where the poles describing the transient response are given by
(53)$$\begin{array}{c} \lambda =\frac{q}{2}\left(-k_{1}\pm \sqrt{k_{1}^{2}-4k_{2}}\right) \end{array}$$

There are only three parameters in the closed-loop system, i.e., two controller gains and one plant gain. From (Equation 52), there are *q* that define the system timescale, $1/k_{2}$, which is the static gain, $\omega_{n}=q\sqrt{k_{2}}$ that denote frequency, and $\xi =\frac{k_{1}}{2\sqrt{k_{2}}}$ that represent the damping coefficient. The parameter $k_{2}$ is required to be set to unity for achieving steady-state reference tracking. This implies that any implementation-related error or deviation in this parameter will be visible in the steady-state error. It follows also that, for an overdamped response, $\xi >1\Rightarrow k_{1}>2$.

### 4.1. Representation with Chemical Reactions

Similar to the previous example, further simplification of the CRN can be achieved through the combination of the integration of the first state with the sum of the feedback contributions and reference. The additional reactions to represent the sum operation suggested in the following studies [12,13,14] can be avoided through this simplification.

By considering the dual-rail representation, the eight catalysis and two annihilation reactions resulting from the CRN are given by
(54)$$R^{+}\overset{q}{\to }R^{+}+X^{+},\;R^{-}\overset{q}{\to }R^{-}+X^{-}$$
(55)$$X^{+}\overset{q}{\to }X^{+}+Y^{+},\;X^{-}\overset{q}{\to }X^{-}+Y^{-}$$
(56)$$X^{+}\overset{qk_{1}}{\to }X^{+}+X^{-},\;X^{-}\overset{qk_{1}}{\to }X^{-}+X^{+}$$
(57)$$Y^{+}\overset{qk_{2}}{\to }Y^{+}+X^{-},\;Y^{-}\overset{qk_{2}}{\to }Y^{-}+X^{+}$$
(58)$$X^{+}+X^{-}\overset{\eta }{\to }Ø,\;Y^{+}+Y^{-}\overset{\eta }{\to }Ø$$

The plant double-integrator is represented by (Equation 54) and (Equation 55), which depicts the chain of two catalysis reactions. The negative gains meanwhile are represented by the reactions in (Equation 56) and (Equation 57), which can be realised through the exchange of contributions between the dual-rail species, as shown in Figure 13a. Lastly, to ensure that the concentrations are maintained at feasible levels, the annihilation reactions are implemented in (Equation 58).

The MAK for the chemical network results in
(59)$${\dot{x}}^{+}=q\left(r^{+}+k_{2}y^{-}+k_{1}x^{-}\right)-\eta x^{+}x^{-}$$
(60)$${\dot{x}}^{-}=q\left(r^{-}+k_{2}y^{+}+k_{1}x^{+}\right)-\eta x^{+}x^{-}$$
(61)$${\dot{y}}^{+}=qx^{+}-\eta y^{+}y^{-}$$
(62)$${\dot{y}}^{-}=qx^{-}-\eta y^{+}y^{-}$$

From the reversed contributions in (Equation 59) and (Equation 60), the negative signs appear in the gains in the I/O dynamics of $\dot{x}={\dot{x}}^{+}-{\dot{x}}^{-}$ and $\dot{y}={\dot{y}}^{+}-{\dot{y}}^{-}$, given by
(63)$$\dot{x}=qr-qk_{2}y-qk_{1}x$$
(64)$$\dot{y}=qx$$

In (Equation 63) to (Equation 64), we recover the linear closed-loop dynamics.

Having the catalysis reactions $X^{\pm }\overset{qk_{1}}{\to }X^{\pm }+X^{\mp }$ crossing the production of each dual-species $X^{\mp }$ together with a fast annihilation $X^{+}+X^{-}\overset{\eta }{\to }Ø$ corresponding to the alternative *catalytic degradation* proposed in [13], with a *self-repressing* gain analogous to having $X^{\pm }\overset{qk_{1}}{\to }Ø$ (for fast $\eta \gg qk_{1}$), hence we can alternatively replace the catalysis in (Equation 56) with degradation reactions. Interestingly, the absence of degradation reactions in the produced species to ensure bounded concentrations and the marginal stability of the stoichiometry in the catalysis reactions can be related to the marginal stability of the integrators in the plant. In this construction, the stabilising feedback gain on *x* can be directly related to degradation reactions on $x^{\pm }$, which introduce a stable pole in the state.

This results in the CRN from Figure 13b with six catalysis, two degradation, and two annihilation reactions given by
(65)$$R^{+}\overset{q}{\to }R^{+}+X^{+},\;\;R^{-}\overset{q}{\to }R^{-}+X^{-}$$
(66)$$X^{+}\overset{q}{\to }X^{+}+Y^{+},\;\;X^{-}\overset{q}{\to }X^{-}+Y^{-}$$
(67)$$X^{+}\overset{qk_{1}}{\to }Ø,\;\;X^{-}\overset{qk_{1}}{\to }Ø$$
(68)$$Y^{+}\overset{qk_{2}}{\to }Y^{+}+X^{-},\;\;Y^{-}\overset{qk_{2}}{\to }Y^{-}+X^{+}$$
(69)$$X^{+}+X^{-}\overset{\eta }{\to }Ø,\;\;Y^{+}+Y^{-}\overset{\eta }{\to }Ø$$

Here, we have a different MAK, i.e.,
(70)$$\begin{array}{cc} {\dot{x}}^{+} & =q\left(r^{+}+k_{2}y^{-}-k_{1}x^{+}\right)-\eta x^{+}x^{-} \end{array}$$
(71)$$\begin{array}{cc} {\dot{x}}^{-} & =q\left(r^{-}+k_{2}y^{+}-k_{1}x^{-}\right)-\eta x^{+}x^{-} \end{array}$$
albeit that $\dot{x}$ has the same I/O dynamics as in (Equation 63). In Figure 14, the comparison of the linear design with the CRN representation is shown. There is a good agreement of the trajectories between the linear control design and the dual-signals obtained from the CRN I/O dynamics, which follows the predefined reference tracking behaviour.

We must keep in mind, however, that the dynamics of the CRN are still a nonlinear system. Writing the MAK of (Equation 65) to (Equation 69) in the new coordinates from Definition 5, we obtain
(72)$$\dot{\bar{x}}=q\left(\bar{r}-k_{2}\bar{y}-k_{1}\bar{x}\right)$$
(73)$$\dot{\bar{y}}=q\bar{x}$$
(74)$$\dot{\hat{x}}=q\left(\hat{r}+k_{2}\hat{y}+k_{1}\hat{x}\right)-\frac{\eta }{2}{\hat{x}}^{2}+\frac{\eta }{2}{\bar{x}}^{2}$$
(75)$$\dot{\hat{y}}=q\hat{x}-\frac{\eta }{2}{\hat{y}}^{2}+\frac{\eta }{2}{\bar{y}}^{2}$$

Again, we have the static state feedback in the I/O system, with nonlinear dynamics unobservable in the representation of the linear feedback system. Although the negative state feedback gains in $\dot{\bar{x}}$ become positive state feedback gains in the dynamics of $\dot{\hat{x}}$, the dynamics in (72) to (75) can be shown to be bounded [23].

### 4.2. Construction with Strand Displacement Reactions

For the DSD representation, the catalysis, degradation, and annihilation reactions are constructed again according to Section 3.2, applying the architectures from Figure 7, Figure 8 and Figure 9, to obtain DSD reactions equivalent to the CRN in (Equation 65) to (Equation 69).

Following the above discussion, the construction in (Equation 65) to (Equation 69) is adhered to for further simplification to the circuit. Here, we replace the catalysis reactions that are responsible for realising the state feedback with the degradation reactions given in (Equation 67) that utilise fewer species and a simpler template complex shown in Figure 8. Nevertheless, solely depending on catalysis (also annihilation) reactions for circuit building may have its own benefit. When proposing a catalytic degradation scheme, the authors in [13] advocated that spatial localisation of the catalysis reaction enables degradation at faster rates.

Moreover, through the Visual DSD simulation and analysis, we observed that the circuit was still functional even with the annihilation reaction $X^{+}+X^{-}\overset{\eta }{\to }Ø$ omitted considering their concentrations stayed low. This is yet another plausible simplification, depending on the experimental setup.

These aforementioned simplifications resulted in the implementation of state feedback having the same complexity level as the integral control problem (i.e., six catalysis, two degradation, and one annihilation reactions), despite the former needing to control a more complex second-order system with marginal stability with more degrees of freedom. In the Visual DSD simulation, there were, respectively, 15 and 20 double-stranded complexes and auxiliary single-stranded species that were initialised at $C_{max}=10^{4}$ nM (the consumption was not reversible and replenished). Furthermore, the maximum toehold binding and unbinding rates were set to $10^{-3}\;{\left(\mathrm{nMs}\right)}^{-1}$ and $0.1\;s^{-1}$, respectively. For the details of all the parameters and the supplied auxiliary strands, see Table 2 and Figure 15, respectively.

In Figure 14, the reference-tracking behaviour of the DSD circuits is shown. We note that there was good agreement between the CRN I/O dynamics with the linear design. The DSD reactions on the other hand had slower transient dynamics. When we compared the concentrations of the CRN with the Visual DSD simulation counterpart (Figure 16), we also observed slower dynamics and lower equilibrium for $Y^{\pm }$ and $X^{\pm }$, respectively, suggesting state *x* was subjected to higher damping.

Nevertheless, the dynamics were similar to the desired design of the state feedback. This observation provided us with another strong indication that we can omit the annihilation reaction for $X^{\pm }$ given its limiting degradation.

## 5. Impacts of Nonlinearities and Uncertainty on Experimental Implementations

The goal of the proposed circuits was to represent two linear feedback systems with chemical reactions, with a number of species and reactions, which can be implemented with currently available capabilities.

Synthetic DNA and strand displacement reactions have been shown to be applicable for biocomputation and implementation of dynamical systems [9,24] and scalable to dozens of distinct molecules [26,32]. Distributed setups can increase even further the number of reactions and species, where the management of the chemical interactions is defined by spatial separation or co-localisation [30], rather than the specificity and orthogonality of the sequences of the programmed toeholds [62]. The constructions proposed in this work relied on systems of DSD reactions of a smaller scale and were based on already implemented schemes such as cooperative hybridisation [32,61].

Despite the obvious potential and capabilities of DNA-based chemistry, from the existing demonstrations of circuits based on DSD reactions, none dealt with the representation of feedback control. The biocomputation of linear feedback systems with nucleic acids remains to be validated experimentally, and beyond the basic validation of the designs, there are additional properties that deserve investigation with an experimental setup.

### 5.1. Positive Equilibrium

It was pointed out in Section 3.1.1 that the linear I/O systems rely on an internal positive and nonlinear representation, albeit unobservable and stable. For a linear IPR, the stability of the IPR implicates a single equilibrium at the origin [59]. In the case of the dual-rail representation applied here and, specifically in the representation of feedback with integral action, previous work from the authors showed that the positive nonlinear dynamics of the CRN realisation can admit an unforced positive equilibrium, which defines the levels of concentration at which the circuit operates and leads to persistent consumption of chemical species [23].

Take again Example 3, assuming that $k_{i}$ was designed for a stable closed-loop system. Then, $\bar{v}$ and $\bar{y}$ in (Equation 32) to (Equation 35) have stable trajectories, and for $r^{\pm }=0$ (no input concentrations), the equilibrium of the I/O system at ${\bar{y}}^{*}=0$ and ${\bar{v}}^{*}=0$ is globally asymptotically stable. For the internal dynamics with $r^{\pm }=0$, the unforced equilibrium conditions result in
(76)$$\begin{array}{cc} 0 & =k_{i}\hat{y}-\frac{\eta }{2}{\hat{v}}^{2}\; \end{array}$$
(77)$$\begin{array}{cc} 0 & =b\hat{v}+a\hat{y}-\frac{\eta }{2}{\hat{y}}^{2} \end{array}$$

Solving for $\hat{y}$, we can write
(78)$$\hat{v}=\frac{\eta }{2b}{\hat{y}}^{2}-\frac{a}{b}\hat{y}\Rightarrow 0=k_{i}\hat{y}-\frac{\eta }{2}{\left(\frac{\eta }{2b}{\hat{y}}^{2}-\frac{a}{b}\hat{y}\right)}^{2}\Rightarrow \left(\frac{\eta^{3}}{8b^{2}}{\hat{y}}^{3}-\frac{a\eta^{2}}{2b^{2}}{\hat{y}}^{2}+\frac{\eta a^{2}}{2b^{2}}\hat{y}-k_{i}\right)\hat{y}=0$$

Besides the trivial solution $\hat{y}=0$, we have from Descartes’ rules of signs at least another equilibrium $\hat{y}={\hat{y}}^{*}>0$, and
(79)$$\hat{y}=0\Rightarrow \hat{v}=0$$
(80)$${\hat{y}}^{*}>0\Rightarrow {\hat{v}}^{*}=+\sqrt{\frac{2k_{i}}{\eta }{\hat{y}}^{*}}>0$$

The reasons for the positive unforced concentrations have been theoretically predicted [23] and can be quickly understood for Examples 3 and 4. Writing the unforced dynamics ($\bar{v}=\bar{y}=r^{\pm }=0$) of the internal dynamics from (Equation 32) to (Equation 35), we have linear and nonlinear components with
(81)$$\begin{array}{c} \left[\begin{array}{c} \dot{\hat{v}}\\ \dot{\hat{y}} \end{array}\right]=\left[\begin{array}{cc} 0 & k_{i}\\ b & -a \end{array}\right]\left[\begin{array}{c} \hat{v}\\ \hat{y} \end{array}\right]-\frac{\eta }{2}\left[\begin{array}{c} {\hat{v}}^{2}\\ {\hat{y}}^{2} \end{array}\right] \end{array}$$

The nonlinear quadratic contribution is always negative and stabilising. However, for the linear term, we have the eigenvalues of the matrix as the solutions of $s\left(s+a\right)-k_{i}b=0$ (which are different from the characteristic polynomial of the I/O system) and
(82)$$\begin{array}{cc} 0 & =s^{2}+as-k_{i}b\Rightarrow \lambda =\frac{1}{2}\left(-a\pm \sqrt{a^{2}+4k_{i}b}\right) \end{array}$$

Due to the positivity of the parameters, we have $k_{i}>0\Rightarrow \lambda =\frac{1}{2}\left(-a+\sqrt{a^{2}+4k_{i}b}\right)>0$. This means the matrix in the internal positive dynamics has modes that push the unforced dynamics away from the origin. The positive eigenvalue is, in fact, the Frobenius eigenvalue, and it can be used to show that the origin is an unstable unforced equilibrium of the MAK [23]. We have the same issue present in Example 4, where, from the dynamics in (72) to (75), we can write the unforced response of the positive dynamics with
(83)$$\begin{array}{c} \left[\begin{array}{c} \dot{\hat{x}}\\ \dot{\hat{y}} \end{array}\right]=q\left[\begin{array}{cc} k_{1} & k_{2}\\ 1 & 0 \end{array}\right]\left[\begin{array}{c} \hat{x}\\ \hat{y} \end{array}\right]-\frac{\eta }{2}\left[\begin{array}{c} {\hat{x}}^{2}\\ {\hat{y}}^{2} \end{array}\right] \end{array}$$
and the eigenvalues of the matrix in the linear term has an unstable positive eigenvector with the associated eigenvalue $\forall_{k_{1},k_{2}>0},\lambda =\frac{q}{2}\left(k_{1}+\sqrt{k_{1}^{2}+4k_{2}}\right)>0$.

The nonlinear quadratic terms ensure the system is bounded, and although for cascaded systems of CRNs without feedback, the annihilation reactions are a practical mechanism to keep the concentrations low, for the representation of feedback, including such reactions may become essential to keep the concentrations bounded [23].

### 5.2. Positive Concentrations and Persistent Consumption

Note that the levels of the operating concentrations and the existence of an unforced equilibrium have direct experimental consequences, such as the continuous depletion of auxiliary strands in the implementation with bimolecular strand displacement reactions.

The histories in Figure 17 show the consumptions of the auxiliary species from the simulation in Visual DSD of the state feedback control in Example 4. The consumed auxiliary species are converted into the outputs of the DSD cascades, which includes the signal species, but also the unreactive waste species that accumulate as the auxiliary species are consumed. Figure 18 shows the histories of the concentrations of the accumulated waste species that result from the DSD implementation in Example 4, where the species Wy and $Wy^{\prime }$ result from the implementation of $Y^{+}+Y^{-}\overset{}{\to }Ø$, while JoinX_1 and $JoinX^{\prime }\_1$ result from sequestering the signal species in $X^{\pm }\overset{}{\to }Ø$. At the end of the DSD cascades for the implementations of the catalysis reactions, there is a waste species: ForkXR_4 and $ForkX^{\prime }R^{\prime }\_4$ result from $X^{\pm }\overset{}{\to }X^{\pm }+R^{\pm }$; ForkYX_4 and $ForkY^{\prime }X^{\prime }\_4$ result from $X^{\pm }\overset{}{\to }X^{\pm }+Y^{\pm }$; $ForkXY^{\prime }\_4$ and $ForkX^{\prime }Y\_4$ result from $Y^{\pm }\overset{}{\to }Y^{\pm }+X^{\mp }$.

The simulation shows the consequences of a positive equilibrium, with the continuous consumption of auxiliary species and the production of waste, even when the output is reaching the steady-state, in Figure 14. The consumption of the auxiliary species should be slow to ensure the approximation of large and constant concentrations of the auxiliary species, during the complete duration of the operation of the DSD circuit. This calls for some care in the parametrisation and experimental setup to ensure the concentrations of auxiliary species remain large during the duration of operation of the circuit or the mechanisms for the replenishment of the auxiliary species and removal of waste species.

The properties of dual-rail implementations of feedback controllers predicted by the above theory have yet to be experimentally tested. The systems proposed in this paper are simple, but they still constitute powerful examples, wherein these theoretical results [22,23] can be tested. The role of annihilation reactions in the implementation of feedback systems, to bound the signal species and to slow down the consumption of auxiliary species and the production of waste species, deserves immediate experimental investigation.

### 5.3. Variability in the Reaction Rates

The underlying assumption of the dual-rail representation of I/O systems is that it relies on perfectly tuned reaction rates. In the derivation of the I/O dynamics in (Equation 30) to (Equation 31), it is assumed in the CRN in (Equation 21) to (Equation 25) that the pairs of equations have exactly the same reaction rates.

However, such an assumption is hindered by the design of affinities based on the complementarity of toeholds, which suffer from granularity and variability [5]. For example, the implementation of the catalysis assumes the designs of different toehold sequences with the same maximum hybridisation (see Remark 1), which we can hope to be similar, but not exactly the same. The differences will change the equilibrium conditions of the reversible reactions and introduce error and uncertainty into the effective rate of output release.

Moreover, besides the hybridisation rates, there is the approximation of large and constant concentrations of the auxiliary species from Assumption 2. Although, theoretically, the concentration of individual auxiliary species can be used to fine-tune the speed of the equivalent unimolecular reaction rates [11], experimental variations will always introduce errors and variability. In Figure 7, we have for catalysis that $k_{i}\propto C_{max}\times c_{i}\times k_{t}$, which is the limiting reaction in (Equation 36) to (Equation 41), and in Figure 8, for degradation, we have $a\propto C_{max}\times c_{a}\times k_{t}$. Hence, the accuracy of the initial concentrations $C_{max}$ impacts the resulting reaction speed.

Consider Example 3, where, despite the system’s simplicity, the physical parameters in the implementation of the degradation reactions impact directly on the dynamic response of the represented I/O system. If we look at the eigenvalues of the closed-loop represented in (Equation 19) given by
(84)$$\begin{array}{c} \lambda =-\frac{a}{2}\pm \frac{1}{2}\sqrt{a^{2}-4k_{i}b} \end{array}$$
we have that, for a high gain $k_{i}>\frac{a^{2}}{4b}$, it results in $ℜ\left\{\lambda \right\}=-\frac{a}{2}$. The real part of the pole depends directly on the implementation of the degradation reactions, and the variability in the designs of the toeholds $c_{a}\times k_{t}$, as well as the impact of the variations of $\left[JoinY\right]\left(t\right)$ during operation, can be observed in the response time.

With a realisation with strand displacement reactions, the parametrisation of the CRN will necessarily depend on several experimental effects, which introduce variability and uncertainty in the reaction rates. The known issues of the dual-rail representation with the mismatching of the reaction rates [11,12] were formally addressed by the author’s theoretical analysis and the characterisation of the asymmetric parametrisation of the dual-rail representations in [22,23].

In particular, the authors showed in [23] that, under parametric variability, the representation of a stable linear model does not necessarily result in a CRN with stable dynamics and the stability of the realisation with DSD reactions. The dual-rail representation relies on the assumption of perfectly symmetrical pairs of chemical reactions, which is broken when considering variability in the parametrisation of the reaction rates. Asymmetric parametrisation of the pairs in the CRNs causes feedback between the input-to-output dynamics and the internal nonlinear positive dynamics of the kinetics, which can lead to unstable feedback within the network, and stability must be analysed for the complete CRN of the dual-rail representation.

However, the emergence of the behaviours predicted by theoretical and numerical analysis and simulation in Visual DSD [22,23] still need validation with actual variability from experimental realisation with DSD reactions.

### 5.4. Robustness

One of the main questions for experimental testing are how the physical parameters impact the reaction rates of the CRN, not only for nominal performance, but also for robustness.

For example, in practice, experimental variability leads to mismatches not only on the representation of the plants with $a^{+}\neq a^{-}$, $b^{+}\neq b^{-}$, and $q^{+}\neq q^{-}$, but also in the representation of the control laws with $k_{i}^{+}\neq k_{i}^{-}$,${\left(qk_{1}\right)}^{+}\neq {\left(qk_{1}\right)}^{-}$, and ${\left(qk_{2}\right)}^{+}\neq {\left(qk_{2}\right)}^{-}$.

The reasons are clearer in the rotated dynamics. Take again Example 3, but now assuming independent reaction rates for all reactions. The MAK then results:(85)$${\dot{v}}^{+}=k_{i}^{+}r^{+}+k_{i}^{-}y^{-}-\eta v^{+}v^{-}$$
(86)$${\dot{y}}^{+}=b^{+}v^{+}-a^{+}y^{+}-\eta y^{+}y^{-}$$
(87)$${\dot{v}}^{-}=k_{i}^{-}r^{-}+k_{i}^{+}y^{+}-\eta v^{+}v^{-}$$
(88)$${\dot{y}}^{-}=b^{-}v^{-}-a^{-}y^{-}-\eta y^{+}y^{-}$$

Expressing the dynamics in the coordinates from Definition 5, we obtain
(89)$$\begin{array}{cc} \dot{\bar{v}} & =k_{i}^{+}r^{+}-k_{i}^{-}r^{-}+k_{i}^{-}y^{-}-k_{i}^{+}y^{+}\; \end{array}$$
(90)$$\begin{array}{cc} \dot{\bar{y}} & =b^{+}v^{+}-b^{-}v^{-}-a^{+}y^{+}+a^{-}y^{-}\; \end{array}$$
(91)$$\begin{array}{cc} \dot{\hat{v}} & =k_{i}^{+}r^{+}+k_{i}^{-}r^{-}+k_{i}^{-}y^{-}+k_{i}^{+}y^{+}-2\eta v^{+}v^{-} \end{array}$$
(92)$$\begin{array}{cc} \dot{\hat{y}} & =b^{+}v^{+}+b^{-}v^{-}+a^{+}y^{+}+a^{-}y^{-}-2\eta y^{+}y^{-} \end{array}$$

**Remark** **2.**
*Note that, for a signal $\bar{y}=y^{+}-y^{-}$ and $\hat{y}=y^{+}+y^{-}$ and applying the same notation to a parameter a such that $\bar{a}=a^{+}-a^{-}$, $\hat{a}=a^{+}+a^{-}$, we have the equivalences*

(93)
$$a^{+}y^{+}-a^{-}y^{-}=\frac{\hat{a}}{2}\bar{y}+\bar{a}\frac{\hat{y}}{2}$$


(94)
$$a^{+}y^{+}+a^{-}y^{-}=\frac{\bar{a}}{2}\bar{y}+\hat{a}\frac{\hat{y}}{2}$$



The correspondences in (93) to (94) allow us to express the rotated dynamics as
(95)$$\begin{array}{cc} \dot{\bar{v}} & =\frac{\hat{k_{i}}}{2}\left(\bar{r}-\bar{y}\right)+\frac{\bar{k_{i}}}{2}\left(\hat{r}-\hat{y}\right)\; \end{array}$$
(96)$$\begin{array}{cc} \dot{\bar{y}} & =\frac{\hat{b}}{2}\bar{v}-\frac{\hat{a}}{2}\bar{y}+\frac{\bar{b}}{2}\hat{v}-\frac{\bar{a}}{2}\hat{y}\; \end{array}$$
(97)$$\begin{array}{cc} \dot{\hat{v}} & =\frac{{\hat{k}}_{i}}{2}\left(\hat{r}+\hat{y}\right)-\frac{\eta }{2}\left({\hat{v}}^{2}-{\bar{v}}^{2}\right)+\frac{{\bar{k}}_{i}}{2}\left(\bar{r}+\bar{y}\right) \end{array}$$
(98)$$\begin{array}{cc} \dot{\hat{y}} & =\frac{\hat{b}}{2}\hat{v}+\frac{\hat{a}}{2}\hat{y}-\frac{\eta }{2}\left({\hat{y}}^{2}-{\bar{y}}^{2}\right)+\frac{\bar{b}}{2}\bar{v}+\frac{\bar{a}}{2}\bar{y}\; \end{array}$$

If the parameters are symmetric, ${\bar{k}}_{i}=\bar{a}=\bar{b}=0$, we recover the desired I/O system decoupled from the positive and nonlinear dynamics. With mismatching reaction rates, we have instead interconnected inputs from the nonlinear to I/O system that change fundamentally the properties of the CRN, and robustness must be addressed for the complete CRN.

Even if the nonlinearities do not participate in the I/O dynamics directly, $\dot{\bar{v}}$ and $\dot{\bar{y}}$ are no longer decoupled from internal positive dynamics. The Assumption 1 of a represented stable I/O system no longer provides guarantees of stability for the interdependent dynamics. Moreover, we can see that, for an unforced response, $\hat{r}=0$, the equilibrium conditions:(99)$$\begin{array}{cc} 0 & =-\frac{\hat{k_{i}}}{2}\bar{y}-\frac{\bar{k_{i}}}{2}\hat{y}\; \end{array}$$
(100)$$\begin{array}{cc} 0 & =\frac{\hat{b}}{2}\bar{v}-\frac{\hat{a}}{2}\bar{y}+\frac{\bar{b}}{2}\hat{v}-\frac{\bar{a}}{2}\hat{y}\; \end{array}$$
(101)$$\begin{array}{cc} 0 & =\frac{{\hat{k}}_{i}}{2}\hat{y}-\frac{\eta }{2}\left({\hat{v}}^{2}-{\bar{v}}^{2}\right)+\frac{{\bar{k}}_{i}}{2}\bar{y}\; \end{array}$$
(102)$$\begin{array}{cc} 0 & =\frac{\hat{b}}{2}\hat{v}+\frac{\hat{a}}{2}\hat{y}-\frac{\eta }{2}\left({\hat{y}}^{2}-{\bar{y}}^{2}\right)+\frac{\bar{b}}{2}\bar{v}+\frac{\bar{a}}{2}\bar{y} \end{array}$$
can have non-zero equilibrium since ${\hat{v}}^{*}>0$ and ${\hat{y}}^{*}>0$ lead to ${\bar{v}}^{*}\neq 0$ and ${\bar{y}}^{*}\neq 0$, with
(103)$$\begin{array}{cc} {\bar{y}}^{*} & =-\frac{\bar{k_{i}}}{\hat{k_{i}}}\hat{y}\; \end{array}$$
(104)$$\begin{array}{cc} {\bar{v}}^{*} & =-\frac{\bar{b}}{\hat{b}}\hat{v}+\hat{y}\left(-\frac{\hat{a}}{\hat{b}}\frac{\bar{k_{i}}}{\hat{k_{i}}}+\frac{\bar{a}}{\hat{b}}\right) \end{array}$$

The experimental variability of the reaction rates entail that, even in the representation of stable linear feedback systems, the error and uncertainty in the parametrisation of the reaction rates, introduced by the implementation with DSD reactions, can lead to a violation of the symmetry assumptions of the dual-rail methodology and result in unstable circuits [23]. The work by the authors in [22] investigated the impact of such variability and mismatch of the reaction rates in the stability of the mass action kinetics of the CRN, based on robust control techniques based on the structured singular value, adapted to address uncertainty, positivity, and nonlinearities. For small variations in the reaction rates and mismatches in the parametrisation of the dual-rail representation, the method can provide a representative and quantified robustness stability margin for the dual-rail-representation of a feedback system.

Therefore, we know how to quantify the robustness of the CRN to inform how much error and uncertainty introduced by the realisation with DSD reaction networks can be tolerated by the chemical representation. However, we lack experimental assessments of such theoretical results, even with the most basic example of the dual-rail representation of feedback.

The proposed systems with reduced complexity rely on a very few parameters, even considering the complete MAK of the CRN. That means that fewer uncertainty intervals need to be characterised and simpler models for robustness analysis can be generated, while still providing examples where theoretical robustness results could be tested experimentally.

### 5.5. Leakage Dynamics

Finally, one of the most-problematic issues in the implementation of DSD networks is the existence of leakage, where the output strands are released even in the absence of input. Leakage without exposed toeholds does exist experimentally, albeit at much lower rates, and strands that are designed to be unreactive will unbind and expose domains and toeholds. For example, leakage in the previously unidirectional in the (Equation 42) reaction can be represented with
(105)$$\begin{array}{ccc} Y+\mathrm{JoinY} & \overset{c_{a}\times k_{t}}{\underset{k_{leak}}{⇌}} & JoinY\_1+dY \end{array}$$

With a finite unbinding rate $k_{leak}$, irreversible reactions in the constructed CRN are actually implemented with reversible cascades of DSD reactions, and such reversibility can change the effective reaction rate and cause the buffering of the input species [8,13].

Moreover, even if the leakage rate can be assumed to be very low $k_{leak}\ll k_{t}\times c_{i}$, the large concentrations of auxiliary strands used in the DSD cascades amplifies the leakage of the output, and its impact on the circuit operation can become relevant [24]. Consider in Example 3 that there are additional and undesired processes that lead to the release of $Y^{\pm }$, where
(106)$$Z^{+}\overset{\rho^{+}}{\to }Z^{+}+Y^{+}$$
(107)$$Z^{-}\overset{\rho^{-}}{\to }Z^{-}+Y^{-}$$

In this case, these reactions model the presence of strands $Z^{\pm }$ that interact with auxiliary species at high concentrations, which can be approximated with unimolecular reactions, where $\rho^{\pm }\approx C_{max}\times k_{leak}$.

**Assumption** **3.**
*Assume that the same leakage processes exist for $Y^{+}$ and $Y^{-}$ and have the same network structure. As the network of unimolecular catalysis and degradation reactions are duplicated, we consider that the leakage processes are also duplicated.*


From the construction methodology of the dual-rail representation, the cascades of DSD reactions for catalysis and degradation have a symmetrical structure. That is, the representation always results in pairs of catalysis and degradation reactions with ideally matching reaction rates. See, for example, Figure 5 and Figure 13, where the symmetry of the construction methodology is clear in the graphs of the networks. With the additional leakage reactions from (106) to (107), we then have in the MAK that
(108)$$\begin{array}{cc} {\dot{v}}^{+} & =k_{i}^{+}r^{+}+k_{i}^{-}y^{-}-\eta v^{+}v^{-}\; \end{array}$$
(109)$$\begin{array}{cc} {\dot{y}}^{+} & =b^{+}v^{+}-a^{+}y^{+}-\eta y^{+}y^{-}+\rho^{+}z^{+} \end{array}$$
(110)$$\begin{array}{cc} {\dot{v}}^{-} & =k_{i}^{-}r^{-}+k_{i}^{+}y^{+}-\eta v^{+}v^{-}\; \end{array}$$
(111)$$\begin{array}{cc} {\dot{y}}^{-} & =b^{-}v^{-}-a^{-}y^{-}-\eta y^{+}y^{-}+\rho^{-}z^{-} \end{array}$$
and
(112)$$\begin{array}{cc} \dot{\bar{v}} & =\frac{\hat{k_{i}}}{2}\left(\bar{r}-\bar{y}\right)+\frac{\bar{k_{i}}}{2}\left(\hat{r}-\hat{y}\right)\; \end{array}$$
(113)$$\begin{array}{cc} \dot{\bar{y}} & =\frac{\hat{b}}{2}\bar{v}-\frac{\hat{a}}{2}\bar{y}+\frac{\bar{b}}{2}\hat{v}-\frac{\bar{a}}{2}\hat{y}+\frac{\hat{\rho }}{2}\bar{z}+\frac{\bar{\rho }}{2}\hat{z}\; \end{array}$$
(114)$$\begin{array}{cc} \dot{\hat{v}} & =\frac{{\hat{k}}_{i}}{2}\left(\hat{r}+\hat{y}\right)-\frac{\eta }{2}\left({\hat{v}}^{2}-{\bar{v}}^{2}\right)+\frac{{\bar{k}}_{i}}{2}\left(\bar{r}+\bar{y}\right)\; \end{array}$$
(115)$$\begin{array}{cc} \dot{\hat{y}} & =\frac{\hat{b}}{2}\hat{v}+\frac{\hat{a}}{2}\hat{y}-\frac{\eta }{2}\left({\hat{y}}^{2}-{\bar{y}}^{2}\right)+\frac{\bar{b}}{2}\bar{v}+\frac{\bar{a}}{2}\bar{y}+\frac{\hat{\rho }}{2}\hat{z}+\frac{\bar{\rho }}{2}\bar{z} \end{array}$$

Even in the absence of input $\hat{r}=0$, the leakage will trigger the production of $\hat{y}$, and indirectly also $\hat{v}$, since in the representation of the feedback, the leaked output species $Y^{\pm }$ will trigger the closed-loop response.

An interesting result that needs experimental investigation is to what extent the symmetry of the dual-rail representation helps with the impact of leakage. For illustration, consider Example 3 with ideal $k_{i}$, *b*, and *a* parameters, but with the existence of leakage with asymmetric rates $\rho^{+}\neq \rho^{-}$. Then
(116)$$\begin{array}{cc} \dot{\bar{v}} & =k_{i}\left(\bar{r}-\bar{y}\right)\; \end{array}$$
(117)$$\begin{array}{cc} \dot{\bar{y}} & =b\bar{v}-a\bar{y}+\frac{\hat{\rho }}{2}\bar{z}+\frac{\bar{\rho }}{2}\hat{z}\; \end{array}$$
(118)$$\begin{array}{cc} \dot{\hat{v}} & =k_{i}\left(\hat{r}+\hat{y}\right)-\frac{\eta }{2}\left({\hat{v}}^{2}-{\bar{v}}^{2}\right)\; \end{array}$$
(119)$$\begin{array}{cc} \dot{\hat{y}} & =b\hat{v}+a\hat{y}-\frac{\eta }{2}\left({\hat{y}}^{2}-{\bar{y}}^{2}\right)+\frac{\hat{\rho }}{2}\hat{z}+\frac{\bar{\rho }}{2}\bar{z} \end{array}$$
and we see that the dynamics of the I/O system depend on the asymmetry of the leakage reaction rate $\bar{\rho }$ and the difference of leaking strands $Z^{\pm }$. If the initial conditions are the same for $Z^{\pm }$ such that $z^{+}\left(t\right)\approx z^{-}\left(t\right)$, then $\hat{\rho }\bar{z}\approx 0$, and even if there is a considerable presence of $Z^{\pm }$, if the leakage process is symmetric such that $\bar{\rho }\approx 0$, then $\bar{\rho }\hat{z}\approx 0$.

We then have that, while the presence of $Z^{\pm }$ can result in serious leakage for each of the components of the signal $y^{\pm }$, with the dual-rail representation of the I/O system, the impact of leakage may be mitigated by reducing the differences between leaked strands and leakage reaction rates. The proposed systems can be used to investigate how the symmetry of the leaking process can be more relevant than the amplitude of the leakage, and if the dual-rail representation can result in systems that are less sensitive to leakage.

## 6. Conclusions

There currently exist mature theoretical frameworks for the design of linear feedback controllers with CRNs. Systematic procedures and software tools provide a translation to equivalent reactions based on nucleic acid chemistry, which should be amenable to experimental implementation. However, the readiness level of this technology has not followed the theoretical developments, and we are lacking experimental validation of such systems.

The viability of the two proposed constructions is increased by the very few DSD reactions and strands necessary for the chemical representation of linear feedback control systems, with the objective of a near-future experimentation and validation of feedback circuits based on toehold-mediated displacement reactions. For the representation of integral control, we propose a representation simpler than found in the literature, by combining in the same pairs of CRNs the operations of gain, subtraction, and integration. With this, we removed from the implementation two catalytic, two degradation, and one annihilation reaction. The representation of the state feedback candidate was simplified by combining the gain, subtraction, and integration of the first state in the same CRNs. Although a double-integrator is not a stable plant, its representation does not need degradation reactions, and with these choices, four catalytic and one annihilation reaction were removed from the representation.

Both representations capture traditional dynamical features of general classes of linear feedback control systems and can be implemented using six catalytic, two degradations, and one or two annihilation reactions. The reduced number of chemical reactions and the size of the DSD networks puts the proposed candidate constructions within current capabilities of experimental implementation.

Although the representations were simplified to fit viable experimentation with current technical capabilities, the examples still represent challenging feedback control systems, and the circuits are interesting for experimental investigation of the dependence of closed-loop dynamics on toehold design and the impact of the annihilation reactions and internal nonlinear positive dynamics on the stability, persistent equilibrium, and concentration levels of the circuit. Furthermore, the proposed systems are rich enough to investigate spurious effects and dynamics present in DSD networks and how much the differential nature of the dual-rail representation may help mitigate experimental perturbations.

## Figures and Tables

**Figure 1 bioengineering-10-00466-f001:**
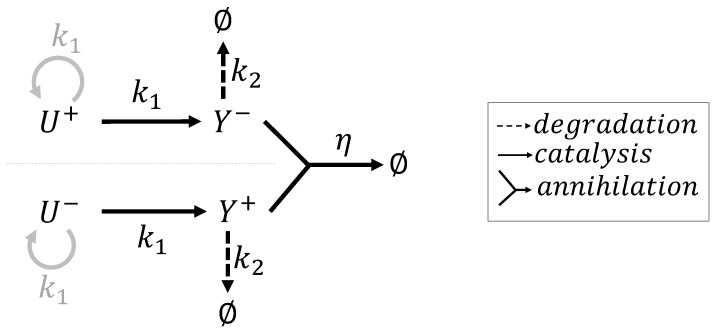
The first-order transfer function described in Example 2 depicted as a network with the dual-rail representation. Here, we have the duplication of unimolecular reactions of degradation and catalysis. In the subsequent graphs, the autocatalysis represented *in grey* will be omitted. Through a bimolecular reaction, the two components of the output signal $y=y^{+}-y^{-}$ annihilate each other.

**Figure 2 bioengineering-10-00466-f002:**
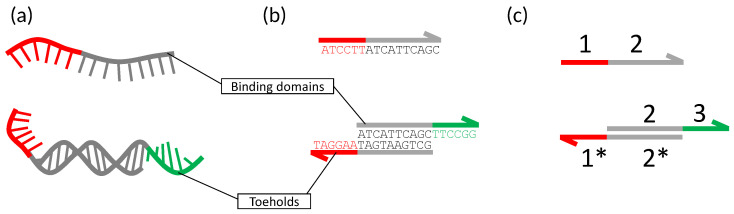
Single- and double-stranded DNA represented in three different manners: (**a**) sugar-phosphate backbones with either forming a helicoidal double structure or exposed sugar bases; (**b**) sequences of bound pairs of complementary sequences of nucleotides; (**c**) numbering the domains, where $2^{*}$ is the complementary sequence to the domain 2.

**Figure 3 bioengineering-10-00466-f003:**
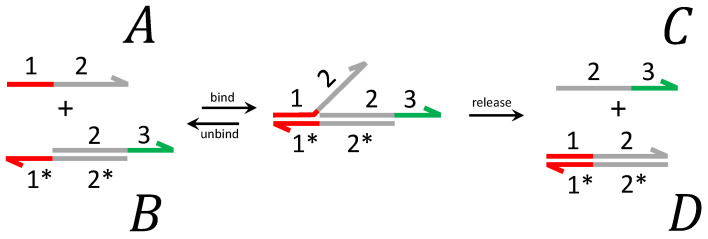
An illustration of a bimolecular DSD reaction of A+B→C+D that involves single- and double-strands. The hybridisation between overhanging complementary toeholds 1 and $1^{*}$ initiates the displacement, resulting in competition for domain $2^{*}$ between the incumbent strand and the incoming strand *A*. The hybridisation carries on till the incumbent is fully displaced, yielding new “species” *C* and *D*. There is an exposed toehold for the single-strand *C*, which can trigger other hybridisation reactions. In view of the absence of exposed toeholds, species *D* is deemed unreactive waste.

**Figure 4 bioengineering-10-00466-f004:**
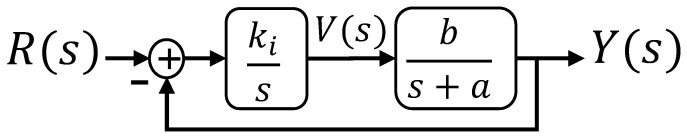
Reference tracking problem with integral control of a stable first-order plant.

**Figure 5 bioengineering-10-00466-f005:**
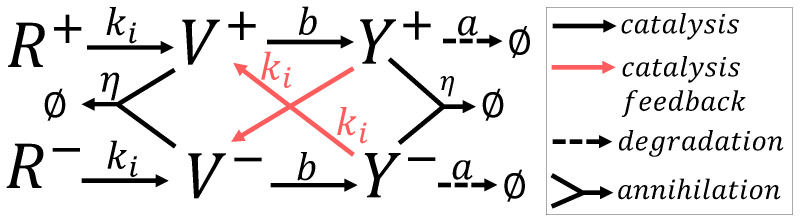
Dual-rail representation for the CRN of the catalysis, degradation, and annihilation reactions for realising integral control employing the MAK.

**Figure 6 bioengineering-10-00466-f006:**
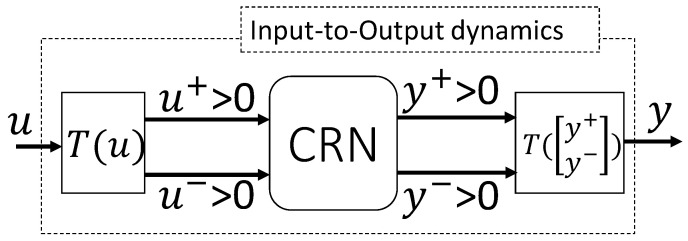
The depiction of input-to-output dynamics of a nonlinear internal positive representation using a linear system, given by the MAK of a CRN.

**Figure 7 bioengineering-10-00466-f007:**
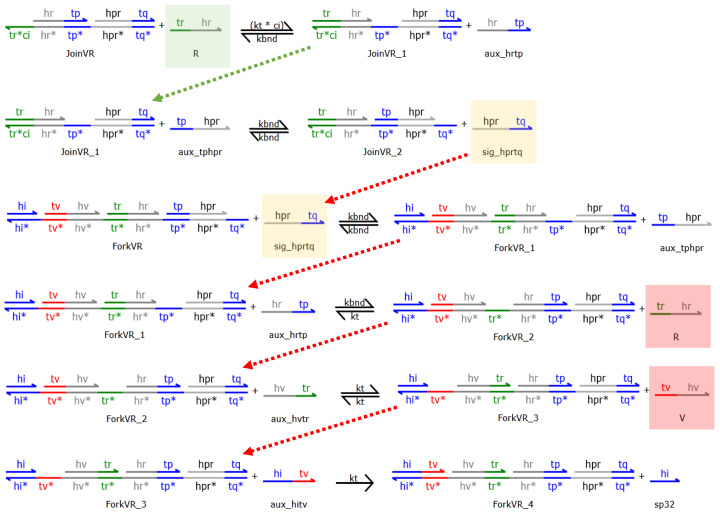
An example of a set of DSD reactions in which their dynamics are equivalent to a catalysis reaction $R\overset{k_{t}\times c_{i}}{\to }R+V$ generated in Visual DSD utilising the Join-Fork templates following [9]. The toehold <tr*ci> in the complex JoinVR contains a complementarity degree to the signal toehold <tr> of $0<c_{i}\leq 1$. This slows down the reaction mediated by <tr> and <tr*ci> to $k_{t}\times c_{i}$, where $k_{t}$ is the maximum binding rate of two complementary toeholds <tr> and <tr*> [13,60].

**Figure 8 bioengineering-10-00466-f008:**
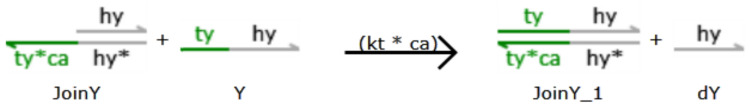
An example of a set of DSD reactions in which their dynamics are equivalent to a degradation reaction $Y\overset{k_{t}\times c_{a},}{\to }Ø$, generated in Visual DSD utilising the Join-Fork templates following [9]. The toehold <ty^∗^ca> has a degree of complementarity of 0 < ca, kt < 1 to the toehold <ty>, effectively slowing down the binding rate from kt to kt×ca.

**Figure 9 bioengineering-10-00466-f009:**
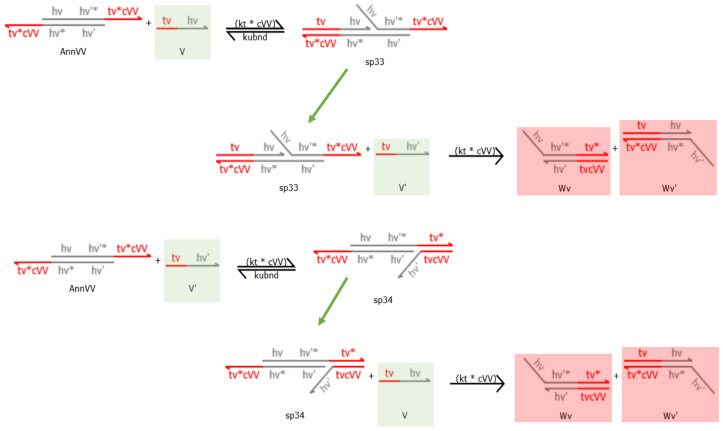
An example of a set of DSD reactions in which their dynamics are equivalent to an annihilation reaction generated in Visual DSD utilising the cooperative hybridisation approach following [32,61].

**Figure 10 bioengineering-10-00466-f010:**
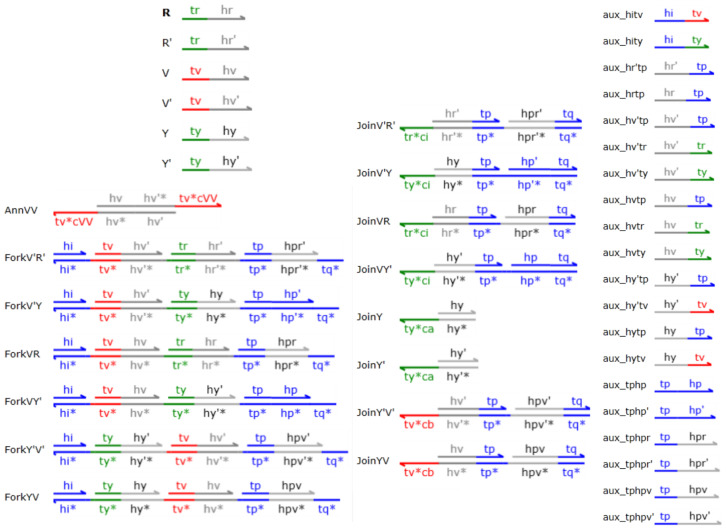
The species that were used to initialise the DSD network for integral feedback control system representation. Here, R, R′, V, V′, and Y, Y′ depict the signal strands. Together with the auxiliary strands, they are initialised at high concentrations.

**Figure 11 bioengineering-10-00466-f011:**
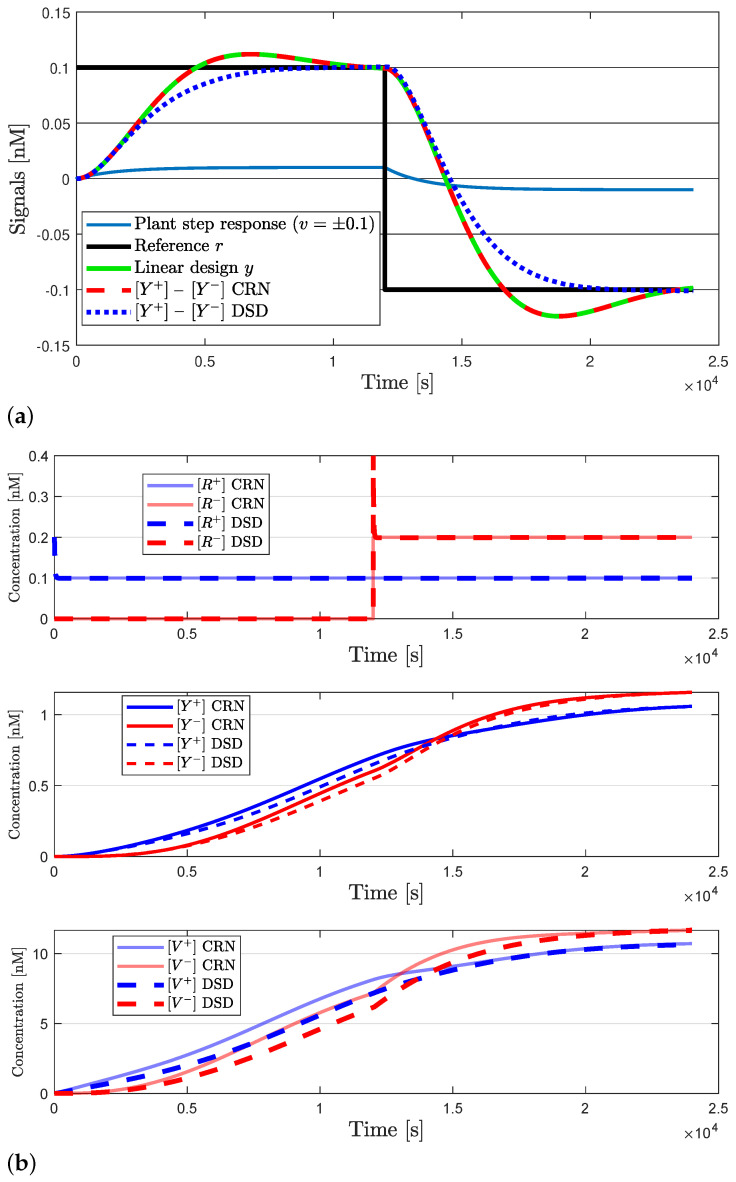
The integral control (Example 3) simulation results: (**a**) output *y* for the step response of the open-loop plant, closed-loop dynamics, the MAK from the CRN and DSD reactions; (**b**) concentrations in both the CRN and DSD representations.

**Figure 12 bioengineering-10-00466-f012:**
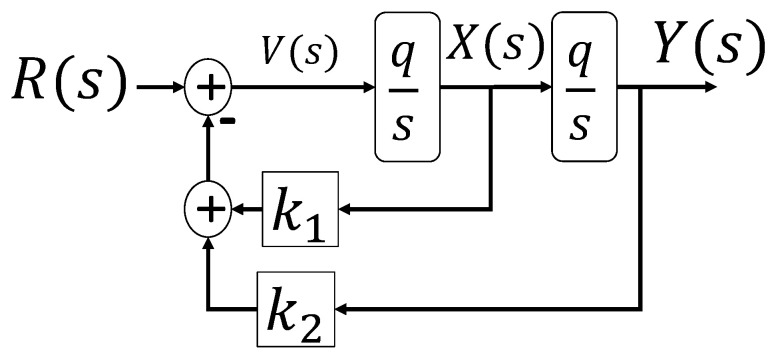
Block diagram representation of a static state feedback controller stabilising a double-integrator plant.

**Figure 13 bioengineering-10-00466-f013:**
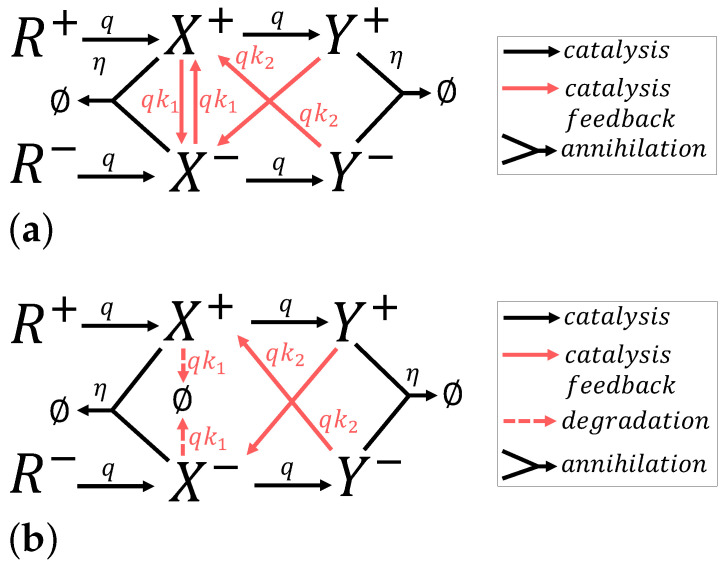
Network of chemical reactions for dual-rail representation of the state feedback control system, using (**a**) *catalytic degradation* or (**b**) degradation reactions.

**Figure 14 bioengineering-10-00466-f014:**
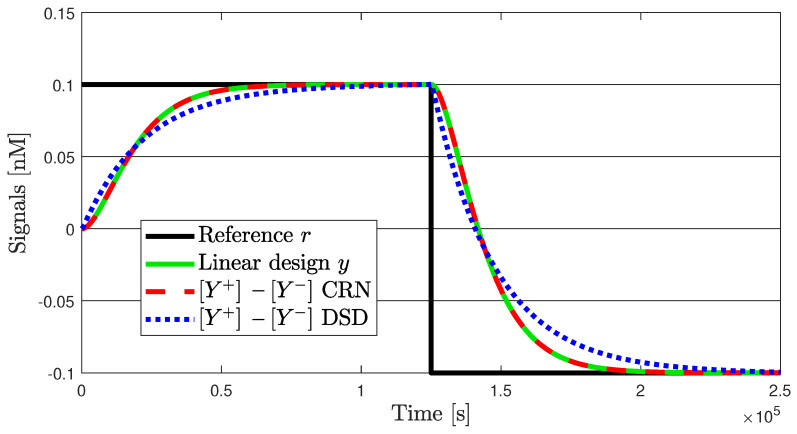
The state feedback control (Example 4) simulation results: steady-state tracking response of the linear design and the CRN and DSD reactions representing the I/O dynamics.

**Figure 15 bioengineering-10-00466-f015:**
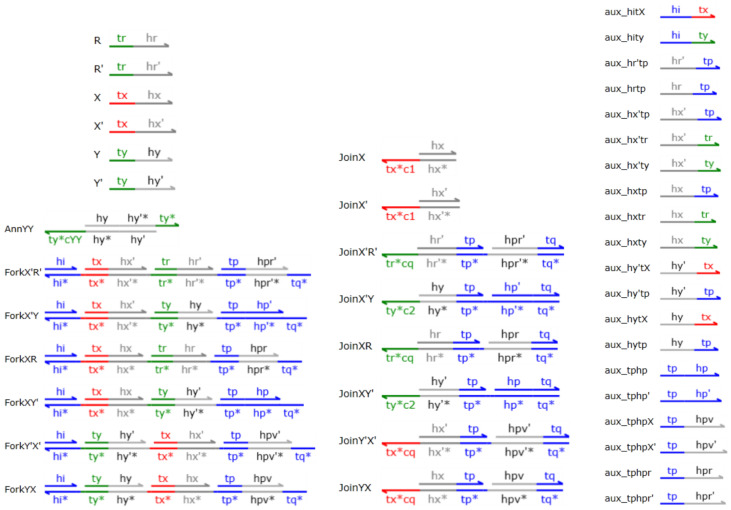
Set of toehold-mediated species present initially for the DSD network representation of the state feedback control of the double-integrator. The strands R, R′, X, X′, and Y, Y′ are the dual-species for the signals. Together with the auxiliary strands, they were initialised at high concentrations.

**Figure 16 bioengineering-10-00466-f016:**
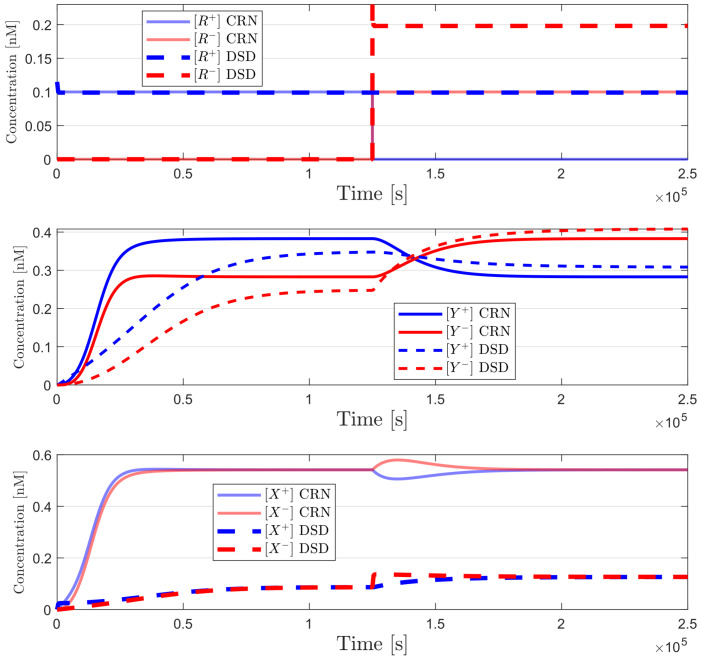
Simulation of the abstract CRN and the DSD implementation for the state feedback control in Example 4. The simulation of the DSD realisation has the output species $Y^{\pm }$ steady-state close to the expected from the CRN representation. $X^{\pm }$, on the other hand, is smaller for the DSD circuit.

**Figure 17 bioengineering-10-00466-f017:**
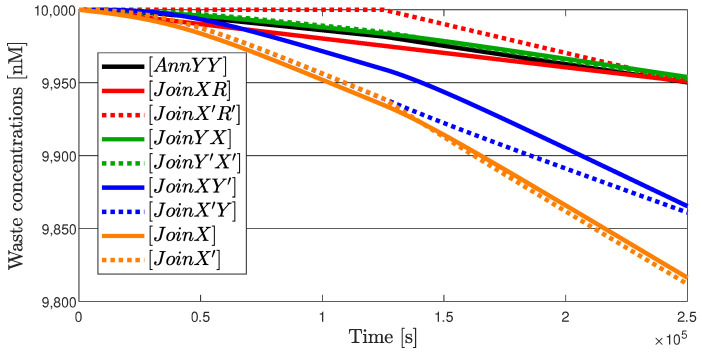
Histories of the consumptions of the auxiliary species from the simulation in Visual DSD of the state feedback control in Example 4, with concentrations remaining close to their initial value of $C_{max}=10^{4}$ nM throughout the operation of the circuit.

**Figure 18 bioengineering-10-00466-f018:**
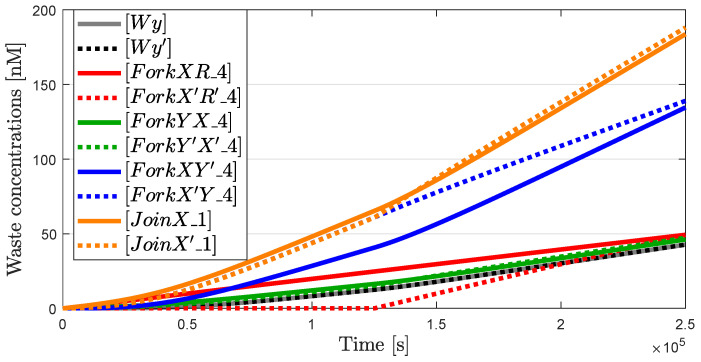
Histories of the concentrations of the waste species from the simulation in Visual DSD of the state feedback control in Example 4: Wy and $Wy^{\prime }$ result from the implementation of $Y^{+}+Y^{-}\overset{}{\to }Ø$; JoinX_1 and $JoinX^{\prime }\_1$ result from implementing $X^{\pm }\overset{}{\to }Ø$; ForkXR_4 and $ForkX^{\prime }R^{\prime }\_4$ result from $X^{\pm }\overset{}{\to }X^{\pm }+R^{\pm }$; ForkYX_4 and $ForkY^{\prime }X^{\prime }\_4$ result from $X^{\pm }\overset{}{\to }X^{\pm }+Y^{\pm }$; $ForkXY^{\prime }\_4$ and $ForkX^{\prime }Y\_4$ result from $Y^{\pm }\overset{}{\to }Y^{\pm }+X^{\mp }$.

**Table 1 bioengineering-10-00466-t001:** The integral control circuit’s parametrisation for Visual DSD simulation.

Variable	Description	Values	Units
$C_{max}$	Template and auxiliary species initial concentrations	$10^{4}$	nM
$k_{bnd}$	Auxiliary strands’ (<tp>,<tq>) toehold maximum binding rate	$10^{-3}$	${\left(\mathrm{nMs}\right)}^{-1}$
$k_{ubnd}$	Unbinding rate	0.1	${\left(s\right)}^{-1}$
$k_{t}$	Signal species’ (<tr>,<ty>,<tv>) toehold maximum binding rate	$10^{-4}$	${\left(\mathrm{nMs}\right)}^{-1}$
$c_{a}$	Degradation of $Y^{\pm }$ toehold complementarity degree	$2.5\times 10^{-3}$	−
$c_{b}$	Plant catalysis reaction toehold complementarity degree	$10^{-3}$	−
$c_{i}$	Feedback catalysis implementing integral control toehold complementarity degree	$5\times 10^{-2}$	−
$c_{VV}$	Annihilation reaction toehold complementarity degree	0.25	−

**Table 2 bioengineering-10-00466-t002:** The state feedback control circuit’s parametrisation for the Visual DSD simulation.

Variable	Description	Vales	Units
$C_{max}$	Template and auxiliary species’ initial concentrations	$10^{4}$	nM
$k_{bnd}$	Auxiliary strands’ (<tp>,<tq>) toehold maximum binding rate	$10^{-3}$	${\left(\mathrm{nMs}\right)}^{-1}$
$k_{ubnd}$	Unbinding rate	0.1	${\left(s\right)}^{-1}$
$k_{t}$	Signal species’ (<tr>,<ty>,<tx>) toehold maximum binding rate	$10^{-4}$	${\left(\mathrm{nMs}\right)}^{-1}$
$c_{q}$	Plant reaction toehold complementarity degree	$8\times 10^{-3}$	−
$c_{1}$	Degradation of $X^{\pm }$ toehold complementarity degree	$8\times 10^{-3}$	−
$c_{2}$	Feedback catalysis of $Y^{\pm }$ toehold complementarity degree	$8\times 10^{-3}$	−
$c_{YY}$	Annihilation reaction toehold complementarity degree	1	−

## Data Availability

The simulation files of this study are available upon reasonable request from the first author, N.M.G.P.

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
