# Peer review of "A Theoretical Framework for Implementable Nucleic Acids Feedback Systems"

_bioengineering, 2023, doi:10.3390/bioengineering10040466_

Round 1
Reviewer 1 Report
Minor revision
This article is an important contribution to find new algorithm treating the feedback of chemical uses.
many approaches were investigated
Results were suported by many analysis
I recommande for authors to add an abreviation list and add more recent work published in 2023 concerning this topic.
with regards
Reviewer 2 Report
Aiming at the problems of backward theory and experimental verification of DNA hybridization, an integral and static negative state feedback for linear controllers was proposed by using chemical reaction network. It's an interesting study and it has theoretical and practical value. Fluent written English, graphic, clear typesetting.
1. Are there page numbers for reference 18? Whether the journal name is an abbreviation or a full name needs to be unified.
2. In Figure 5, are the two ki the same? Obviously, ki values of positive and negative feedback are not the same, requiring different mathematical symbols.
3. There are too many formulas. Please check them carefully.
Reviewer 3 Report
Synthetic biology in the last years has acquired high interest for its peculiar properties, which allow its applications in different fields. This manuscript provides a chemical reaction network description, that represents support for the experimental reactions.
The authors provided a detailed description of the systems in the different conditions, including the variability in the reaction rates. The impacts of the non-linearity and uncertainty on experimental implementation have been described.
Also, the theoretical data on systems’ robustness have been considered. However, experimental validation is necessary to confirm the reported data.
Most probably the authors have already considered a follow-up paper where the theoretical data will be complemented by the experimental data to confirm the claims.
The entire study has been well and carefully executed, the applied methods seem to be state-of-the-art, and the results are presented and comprehensible even for a non-expert in computational simulation techniques.
The Author made appropriate references.
The length of the manuscript is fine.
Without any doubt, the conclusions drawn are interesting and likely to interest abroad readership.
For these reasons, I recommend the publication of this piece in the Bioengineering Journal in its present form.
